# Optimal Computed Tomographic Arthrography Protocol for Stifle Ligamentous Structure and Menisci in Dogs

**DOI:** 10.3390/ani14223334

**Published:** 2024-11-19

**Authors:** Jiwon Yoon, Gunha Hwang, Soyon An, Young Joo Kim, Tae Sung Hwang, Hee Chun Lee

**Affiliations:** 1Institute of Animal Medicine, College of Veterinary Medicine, Gyeongsang National University, Jinju 52828, Republic of Korea; nlz6ha94@gmail.com (J.Y.); hgh3634@gmail.com (G.H.); rapunzel9367@gmail.com (S.A.); 2College of Veterinary Medicine, Western University of Health Sciences, Pomona, CA 91766, USA; yjkim@westernu.edu

**Keywords:** computed tomography, arthrography, stifle joint, ligamentous structure, dogs

## Abstract

This study evaluates the optimization of computed tomographic arthrography (CTA) protocols for delineating ligamentous structures and menisci in the stifle joints of dogs. The motivation stems from existing diagnostic challenges, such as inaccuracies in physical and radiographic assessments often observed in conditions like cranial cruciate ligament rupture and meniscal injuries. Various protocols involving different concentrations and volumes of the contrast medium iohexol were tested to determine the most effective method for enhancing visualization of these structures. The findings reveal that a contrast medium concentration of 100 mgI/mL, administered at volumes of 0.3 or 0.4 mL/kg, significantly improves the clarity and detail of imaging, providing a more accurate diagnostic tool for veterinary practitioners. This optimized protocol could significantly enhance clinical outcomes by facilitating more precise diagnosis of joint diseases in dogs.

## 1. Introduction

Joint instability, particularly in the stifle joint, is a common joint disease in dogs. One of the common causes of hindlimb lameness is cranial cruciate ligament rupture (CCLR) [1,2,3]. To accurately diagnose problems in the stifle joint, a detailed evaluation of the joints, ligaments, and associated structures is necessary. However, there are limitations when conducting these assessments through physical examination and radiographic evaluation. For example, during a physical examination, a conscious patient may show muscle contracture due to apprehension or soft tissue proliferation from chronic injury. These conditions, including partial CCLR, often result in false-negative results [4,5,6]. Additionally, radiographic evaluation can be challenging because dogs with and without meniscal injuries may show minimal differences [7].

Several diagnostic methods have been established for effectively detecting diseases of the stifle joint, such as CCLR and meniscus injury. These methods include ultrasound examination [8,9,10,11,12], computed tomographic arthrography (CTA) [12,13,14,15,16,17,18], magnetic resonance imaging (MRI), MRI arthrography [19,20,21], and arthroscopy [22,23,24]. Among these methods, CTA is particularly advantageous in terms of cost, efficiency, and ease of use. Unlike ultrasonography, it requires less operator-dependent skill [8,9,11]. Compared to MRI, CTA is more economical, quicker, and can be performed with sedation alone, even in areas where surgical hardware may compromise MRI clarity [15,17,19,21,25]. Additionally, CTA’s minimally invasive approach and reliable imaging offer benefits over arthroscopy, which may miss internal meniscal lesions [16,17,24].

There are various documented CTA protocols specifically designed to assess the stifle joint in dogs. One protocol involves injecting iohexol at a concentration of 150 mgI/mL using a dosage of 0.3 mL for each centimeter of the joint’s medial–lateral width when dealing with partial cranial cruciate ligament ruptures [14]. Another approach utilizes up to 5 mL of iohexol at 175 mgI/mL to diagnose medial meniscal lesions [15]. In a different study, injury detection in the medial meniscus of dogs with cranial cruciate ligament insufficiency was explored using a volume of 0.2 mL/kg of sodium and meglumine amidotrizoate at a concentration of 185 mgI/mL [18]. The effectiveness of arthrography heavily relies on the concentration of the contrast agent. A low concentration runs the risk of dilution within the body and inadequate tissue contrast, while a high concentration may result in better tissue differentiation but increase the occurrence of bloom artifacts [13]. Additionally, an insufficient volume of contrast agent may not adequately cover the structures, compromising the clarity of their contours [12]. Therefore, it is crucial to tailor an optimized CTA protocol to enhance the visualization of ligamentous structures and menisci in the stifle joint. Surprisingly, there has been no reported research on this optimization. Hence, the purpose of this study is to optimize the concentration and injection volume of the contrast medium in order to improve visualization for the purpose of evaluating the stifle joint’s ligamentous structures and menisci.

## 2. Materials and Methods

### 2.1. Preparation and General Anesthesia of the Experimental Animals

All procedures and animal care were approved by Institutional Animal Care and Gyeongsang National University (GNU-220929-D0116) user committees. Six healthy beagle dogs (mean weight: 11.1 kg, range: 9.7–12.5 kg; mean age: 6 years; 6 castrated males) were included in this study. Prior to receiving general anesthesia, all dogs underwent detailed physical examinations, hematobiochemical analysis, and thoracic and musculoskeletal radiographic imaging. No clinical symptoms related to the musculoskeletal system were observed, and no abnormalities were detected. To minimize the effects of the previous CT arthrography and to consider the patient’s condition, a two-week interval was maintained between each CT arthrography. Anesthesia was induced using alfaxalone (2 mg/kg, IV, Alfaxan inj^®^, Careside. Co., Ltd., Gyeonggi-Do, Republic of Korea) and maintained with isoflurane (Irfan^®^, Hana Pharm. Co., Ltd., Gyeonggi-Do, Republic of Korea) administered in oxygen (2.0 L/min) via endotracheal intubation. Throughout the procedures, electrocardiography (ECG), oxygen saturation, and breathing were continuously monitored.

### 2.2. Protocols of CT

We used a 160-multislice CT scanner (Aquilion lightning 160, Canon Medical Systems, Otawara, Japan) to collect all imaging data. The dog was positioned on the scanning table in dorsal recumbency, with the hindlimbs extended caudally. To achieve approximately 135° of flexion, we used specially designed bone alignment guides made of wood. We performed helical CT scanning without the use of contrast agents, covering the anatomical span from the coxofemoral to the tarsal joint, with a focus on the stifle joint. The scanning parameters were set to a voltage of 120 kVp, a variable mAs, a slice thickness of 0.5 mm, a scan field of view of 320 mm, a rotation time of 0.75 s, and a pitch factor of 0.637.

### 2.3. CT Anthrography

A total of 54 procedures were performed in six dogs, with each dog undergoing the nine CT arthrography protocols using various concentrations and volumes of contrast agents. A 50 mgI/mL concentration of contrast medium was injected at volumes of 0.2 mL/kg, 0.3 mL/kg, and 0.4 mL/kg (protocols S1, S2, and S3). A 100 mgI/mL concentration of contrast medium was injected at volumes of 0.2 mL/kg, 0.3 mL/kg, and 0.4 mL/kg (protocols M1, M2, and M3). A 150 mgI/mL concentration of contrast medium was injected at volumes of 0.2 mL/kg, 0.3 mL/kg, and 0.4 mL/kg (protocols L1, L2, and L3). All contrast agents were prepared by diluting iohexol (Omnipaque 300 mgI/mL, GE Healthcare, Denmark) with normal saline (Table 1).

All arthrograms were performed by a single experienced veterinarian (JY) with over two years of experience in radiology. The procedure was only performed on the left leg. First, the stifle joint area was shaved and cleaned in a sterile manner. Then, a pre-contrast scan was performed, and a 22-gauge needle was inserted medially into the patellar tendon. Synovial fluid was drawn from the joint to confirm the needle’s position, and CT arthrography was performed following the established protocols. The joint was moved repeatedly to ensure the contrast medium was evenly distributed throughout the joint space. After that, the limb was repositioned on the CT table as before, and the imaging protocol was repeated. Once the CT imaging was completed, the contrast agent was removed through arthrocentesis to reduce swelling, pain, and potential impact on subsequent arthrography procedures.

### 2.4. Image Analysis

All images were reconstructed in the transverse plane using a bone algorithm, with an image thickness of 0.5 mm and a scan field of view of 80 mm. A total of four veterinarians who each had more than 3 years of experience in radiology conducted this study. Before interpreting the images, the veterinarians received instructional materials that included a reference paper [13] and six normal canine CT stifle arthrograms. The veterinarians used a DICOM viewer (RadiAnt DICOM viewer, version 2020.2; Medixant Inc., Poznan, Poland) for the training and evaluation of CT arthrography. To assess the transverse images, the veterinarians used the multiplanar reconstruction (MPR) function of the viewer to convert them into the sagittal and coronal planes. Following the procedures outlined in a previous paper, the veterinarians used MPR to align the scans. This ensured a standard orientation where the transverse plane was parallel to the tibial plateau, the coronal plane was orthogonal to it, and the sagittal plane was set at a right angle to both the transverse and coronal planes [15].

The following anatomic structures were evaluated in each scan (Figure 1): cranial cruciate ligament (CrCL), caudal cruciate ligament (CdCL), medial meniscus (MM), lateral meniscus (LM), meniscofemoral ligament (MFL), medial cranial meniscotibial ligament (MCrMTL), lateral cranial meniscotibial ligament (LCdMTL), medial caudal meniscotibial ligament (MCdMTL), lateral caudal meniscotibial ligament (LCdMTL), and intermeniscal ligament (IML). The scoring for the visualization of stifle ligamentous structures and menisci was based on the following criteria: 0—the structure was not visible; 1—the structure was detectable but poorly visualized, identified by its location and filling defect rather than its margins or shape; 2—the structure was visualized by its location, shape, and filling defect, though margins were not clearly delineated due to underfilling or artifacts; and 3—the structure was well-visualized and sharply delineated, characterized by its location, shape, filling defect, and margins (Table 2 and Figure 2).

### 2.5. Statistical Analysis

All statistical analyses were conducted using SPSS software (SPSS for Windows, Release 27.0, standard version, SPSS Inc., Chicago, IL, USA). The intra- and inter-observer agreement among the four observers was assessed using the Fleiss’ kappa with a 95 percent confidence interval. The visualization of the stifle ligamentous structures and menisci was statistically analyzed using the Kruskal–Wallis test. The Kruskal–Wallis test was employed to compare each group, and post hoc Mann–Whitney U tests were carried out to compare the concentration and injection volume of the contrast medium. For the statistical tests, a *p*-value of less than 0.05 was considered significant for the Kruskal–Wallis test, and a value less than 0.0013 was considered significant for the Mann–Whitney U test.

## 3. Results

### 3.1. Intraclass and Inter-Observer Correlation Coefficient

A reliability assessment was conducted to evaluate the visualization scores of the ligamentous structures and menisci in the stifle (Table 3). The interpretation of the kappa (*k*) values followed standard guidelines: poor agreement (<0), slight agreement (0.01–0.20), fair agreement (0.21–0.40), moderate agreement (0.41–0.60), substantial agreement (0.61–0.80), and almost perfect agreement (0.81–1.00).

For intra-observer agreement, the cranial cruciate ligament (CrCL), caudal cruciate ligament (CdCL), medial meniscus (MM), lateral meniscus (LM), and meniscofemoral ligament (MFL) achieved substantial agreement, with *k* values ranging between 0.6 and 0.8. The medial cranial meniscotibial ligament (MCrMTL), lateral cranial meniscotibial ligament (LCrMTL), medial caudal meniscotibial ligament (MCdMTL), lateral caudal meniscotibial ligament (LCdMTL), and intermeniscal ligament (IML) demonstrated moderate agreement, with *k* values ranging between 0.401 and 0.6.

For inter-observer agreement, the CrCL, CdCL, MM, and LM showed substantial agreement, with *k* values between 0.6 and 0.8. The MFL, MCrMTL, LCrMTL, MCdMTL, LCdMTL, and IML exhibited moderate agreement, with *k* values ranging from 0.401 to 0.6.

These findings indicate that both the intra- and inter-observer reliability were acceptable, with substantial to moderate agreement depending on the specific ligamentous structure or meniscus assessed.

### 3.2. Qualitative Image Analysis of Visualization

A detailed qualitative analysis of the visualization results for the stifle ligamentous structures and menisci is presented in Figure 3. It was found that the overall visualization of the stifle ligamentous structures and menisci in groups M2 and M3 was notably superior, receiving high scores.

Specifically, the visualization of the cranial and caudal cruciate ligaments was significantly better in groups M2 and M3 compared to groups S1 and L3. Additionally, the medial meniscus was less clearly visualized in groups S1 and S2, while the lateral meniscus was not as effectively seen in group S1 compared to groups M2, M3, L1, L2, and L3. Furthermore, the meniscofemoral ligament was significantly more visible in groups S3, M2, and M3 than in group L3. The visualization of the medial cranial meniscotibial ligament was found to be better in groups M2 and M3 compared to groups S1 and L3. A similar trend was observed for the lateral cranial meniscotibial ligament, with groups M2, M3, and L1 showing superior visualization compared to groups S1 and S2. Additionally, the medial and lateral meniscotibial ligaments were significantly better visualized in groups M2 and M3 compared to group S1. Moreover, the lateral meniscotibial ligament demonstrated significant improvements in visualization in groups M2 and M3 compared to group S2.

Finally, the visualization of the intermeniscal ligament was significantly better in groups M2 and M3 compared to groups S1, L2, and L3. Although there was no statistical significance, there was a general trend where the visualization of most ligamentous structures and menisci was also good in groups S3, M1, and L1, in addition to groups M2 and M3.

## 4. Discussion

There are several critical factors in CTA that contribute to the quality of imaging outcomes. These factors include the contrast medium, injection technique, patient positioning, scanning parameters, and the radiologist’s interpretation experience [16,18,26,27,28]. Among these factors, the contrast medium is particularly important in determining the quality of the images. The concentration of the contrast medium plays a crucial role in enhancing the contrast of intra-articular ligaments [13,29,30,31]. If the volume of the contrast medium is insufficient, it may not fully capture the intra-articular ligaments, resulting in poorly defined contours [12]. On the other hand, if the volume is excessive, it can obscure these ligaments, making their accurate evaluation challenging [15]. Therefore, it is essential to use the contrast medium appropriately in order to enhance image quality. This can be easily adjusted in clinical practice, along with patient positioning.

In a cadaver study on CT arthrography of the stifle joint, researchers found that a 1.5 mL dose of contrast agent, with a concentration of 150 mgI/mL, was the most effective in clearly distinguishing the structures of the stifle ligaments. This concentration was compared to others, such as 75, 150, and 300 mgI/mL [13]. However, this study did not consider that ligament visibility may improve if the volume is adjusted based on the concentration, as it consistently used a volume of 1.5 mL regardless of concentration. Moreover, concentrations of 300 and 150 mgI/mL are relatively high compared to other studies on joint arthrography, highlighting the need for further research on lower contrast agent concentrations. For evaluating the shoulder in dogs, a concentration range of 60–80 mgI/mL, from a broader spectrum of 10–100 mgI/mL, was deemed appropriate [31]. In this study, groups M2 (100 mgI/mL, 0.3 mL/kg) and M3 (100 mgI/mL, 0.4 mL/kg) showed significantly enhanced visualization of all stifle joint ligaments and menisci. However, these results were not significantly different from those in group M1, which had the same concentration at 0.2 mL/kg. Therefore, it was concluded that performing arthrography with a contrast agent concentration of 100 mgI/mL and volumes of 0.2, 0.3, or 0.4 mL/kg would likely result in good visualization of the stifle joint ligaments and menisci.

It has been observed that using a higher concentration of contrast medium for arthrography increases tissue contrast. However, this can also lead to an escalation in beam hardening artifacts. In addition, hypertonic iodinated solutions have been associated with post-procedural swelling and pain [29]. High-concentration contrast agents can alter the osmotic pressure within cartilage tissue, potentially increasing the water content and inducing swelling. Additionally, high-concentration iodine contrast agents have an increased propensity to diffuse into damaged cartilage regions by following micro-cracks on the surface. This enhanced penetration improves lesion visualization, but also elevates the risk of inflammation and pain due to localized edema and tissue reaction in structurally compromised cartilage [32]. Reports have indicated that certain concentrations of contrast medium can cause artifacts that obscure the visualization of ligament margins in the canine stifle [13] and shoulder [31] during CT arthrography. In this study, it was observed that in group L3 (150 mgI/mL, 0.4 mL/kg), the occurrence of artifacts significantly obscured the margins of the CrCL, CdCL, MFL, MCrMTL, and IML. Similarly, in group L2 (150 mgI/mL, 0.3 mL/kg), a similar concentration resulted in significant obscuration of MCrMTL and IML margins. However, there was no significant difference between group L1 (150 mgI/mL, 0.2 mL/kg) and group M (100 mgI/mL), suggesting that higher concentrations of contrast medium (150 mgI/mL) used at larger volumes (0.3–0.4 mL/kg) are not suitable for visualizing CrCL, CdCL, MFL, MCrMTL, and IML during arthrography. Consequently, it is considered that using a smaller volume (0.2 mL/kg) may lead to better visualization of all stifle joint ligament structures and menisci.

Low concentrations of contrast medium can dilute within the body, which can result in inadequate contrast differentiation between tissues [29]. In vitro studies indicate that using lower concentrations of iodinated contrast may help reduce viscosities and potentially improve the visualization of small intra-articular structural irregularities [33]. In dogs, a concentration of contrast medium at 75 mgI/mL with a volume of 1.5 mL is too diluted to properly evaluate stifle joint ligaments [13]. In this study, group S1 (50 mgI/mL, 0.2 mL/kg) did not provide significantly improved visibility of the stifle joint ligaments and menisci, suggesting that the contrast was insufficient. Similarly, in group S2 (50 mgI/mL, 0.3 mL/kg), the visibility of MM, LCrMTL, and LCdMTL did not show a significant improvement. However, there was no notable difference between group S3 (50 mgI/mL, 0.4 mL/kg) and group M (100 mgI/mL), indicating that performing arthrography with a lower concentration of contrast medium (50 mgI/mL) at a lower volume (0.2 mL/kg) may lead to excessive dilution and inadequate tissue contrast. Therefore, a greater volume (0.3–0.4 mL/kg) of contrast media, with a lower concentration, is believed to be more effective in improving the visibility of all ligament structures and menisci in the stifle joint.

The medial meniscus is firmly attached to the tibial posterior intercondylar area and is located in front of the caudal cruciate ligament. Unlike the medial meniscus, the lateral meniscus is not rigidly connected to the collateral ligament, allowing for greater mobility. Its flexible peripheral attachment is interrupted by the popliteus tendon [34]. In this study, CTA showed that the menisci were well visualized in groups M2 (100 mgI/mL, 0.3 mL/kg) and M3 (100 mgI/mL, 0.4 mL/kg), as well as in groups L1 (150 mgI/mL, 0.2 mL/kg), L2 (150 mgI/mL, 0.3 mL/kg), and L3 (150 mgI/mL, 0.4 mL/kg), unlike other ligament structures in the stifle joint. This improved visibility is likely because the menisci occupy a larger surface area within the joint, which prevents them from being obscured by artifacts that are often associated with higher concentrations of contrast agents. Additionally, in group S2 (50 mgI/mL, 0.3 mL/kg), the medial meniscus was not as easily seen as the lateral meniscus. This could be because the medial meniscus is smaller and tightly attached to the joint capsule and medial collateral ligament. These factors can make it difficult to distinguish the meniscus clearly when using lower concentrations of contrast medium or an insufficient volume for proper distribution within the joint. Therefore, it is recommended to use a contrast medium with a volume of 0.4 mL/kg and a concentration of 50 mgI/mL, or to consider using a concentration higher than 100 mgI/mL, for better visualization of structures closely connected to the joint capsule and occupying a significant area.

The ligaments MCrMTL and IML could only be clearly seen in groups M2 (100 mgI/mL, 0.3 mL/kg) and M3 (100 mgI/mL, 0.4 mL/kg), unlike other ligaments in the stifle joint in this study. This finding is important because these ligaments are small and complex. However, because they are close to the medial meniscus and occupy a limited area within the joint, using a contrast concentration of 50 mgI/mL in the stifle joint did not provide enough contrast to clearly differentiate the tissues, making visualization difficult. Additionally, a concentration of 150 mgI/mL was considered too high, as it could potentially obscure the ligament margins due to their small size and the presence of artifacts. Consequently, to effectively visualize the intricate ligaments within the joint capsule, it has been determined that a contrast agent concentration of 100 mgI/mL with a volume exceeding 0.3 mL/kg is appropriate.

For intra- and inter-observer agreement, structures that are well visualized by CTA, such as the CrCL, CdCL, MM, and LM, demonstrated *k* values ranging from 0.6 to 0.8, indicating substantial agreement. However, the remaining structures generally showed *k* values within the range of 0.41–0.60, corresponding to moderate agreement. This lower level of agreement is likely due to the difficulty of clearly identifying these structures, which affected the evaluators’ consistency.

Additionally, as a characteristic of Fleiss’ kappa analysis, it adjusts for agreement occurring by chance, which can lead to lower *k* values even when the actual data agreement is high. This suggests that there is a potential for underestimation in the agreement results.

In this study, the dog was placed in dorsal recumbency and the hindlimbs were extended caudally, with the stifle positioned at approximately 135° of flexion. The dorsal recumbency position was chosen because it allows for an extended stifle orientation, aligning the tibial plateau parallel to the gantry. This alignment makes it easier to position the patient’s back parallel to the gantry [12]. It was confirmed that 135° represents the typical stifle angle in standing dogs [35].

This study has a few limitations. First, we only conducted the experiment on one breed, the beagle, which limited the diversity of our sample. The number of animals we used was small and their body weights were similar, so we could not assess a wider range of weights. Additionally, due to the small number of animals, there is a possibility that statistical Type I and Type II errors may have occurred, which underscores the need for further studies involving a larger number of animals. This limitation is especially important for larger breeds, as their joint capsules may vary significantly in size due to their higher body weight. Therefore, further research is needed to confirm whether our findings can be applied to larger breeds. Secondly, the evaluations were conducted only on healthy, normal dogs. It is expected that the evaluation protocol may need to be modified for dogs with diseases, as joint effusion or inflammatory factors could impact the results. Therefore, further research is needed for patients with medical conditions. Lastly, because imaging at flexion angles was not feasible, a comparative analysis using different extension positions could not be performed. This limitation means that the ideal positioning for CTA remains unverified, highlighting the need for additional research to determine the optimal imaging posture for CTA.

## 5. Conclusions

Identifying the optimal concentration and volume of contrast agent is essential to enhance ligament visibility in CT arthrography while minimizing adverse effects. We recommend that for CT arthrography, the most effective way to visualize intra-articular structures is by administering a contrast medium concentration of 100 mgI/mL at a dose of either 0.3 mL/kg or 0.4 mL/kg. If higher concentrations of contrast agent, like 150 mgI/mL, are used, it may be beneficial to use lower doses, such as 0.2 mL/kg. Conversely, if lower concentrations of contrast agent, like 50 mgI/mL, are used, larger doses, such as 0.4 mL/kg, may be necessary to improve visualization of the stifle joint ligaments and menisci.

## Figures and Tables

**Figure 1 animals-14-03334-f001:**
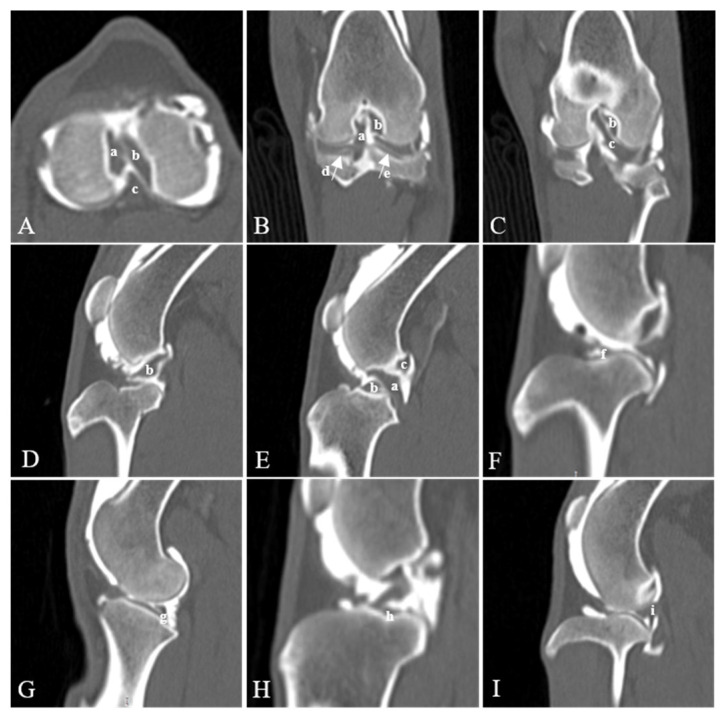
Visualization of stifle joint ligamentous structures and menisci in CT arthrography. Axial (**A**), coronal (**B**,**C**), and sagittal (**D**–**I**) planes. Cranial cruciate ligament (a), caudal cruciate ligament (b), medial meniscus (c), lateral meniscus (d), meniscofemoral ligament (e), medial cranial me-niscotibial ligament (f), lateral cranial meniscotibial ligament (g), medial caudal meniscotibial ligament (h), lateral caudal meniscotibial ligament (i).

**Figure 2 animals-14-03334-f002:**
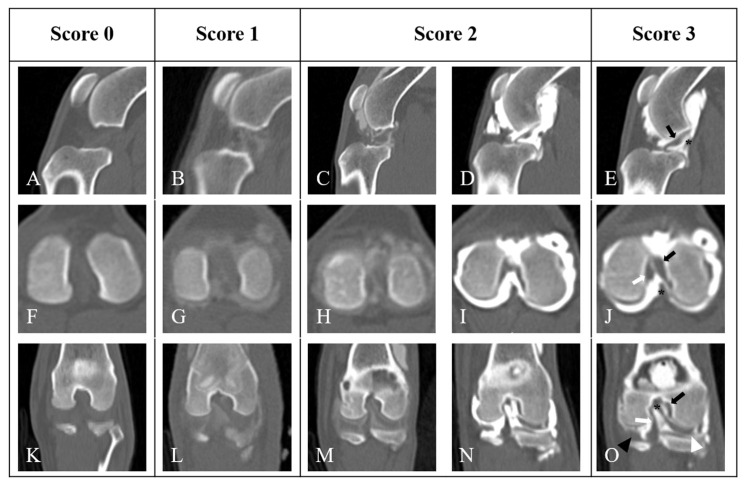
Standard of scoring for visualization of stifle joint ligamentous structures and menisci in CT arthrography. Sagittal (**A**–**E**), Axial (**F**–**J**) and coronal (**K**–**O**) planes. Scores are assigned as follows: Score = 0 (non-visualized (**A**,**F**,**K**)), Score = 1 (poorly visualized (**B**,**G**,**L**)), Score = 2 (visualized with minor limitations such as underfilling (**C**,**H**,**M**) or with artifacts (**D**,**I**,**N**)), and Score = 3 (well-visualized with clear detail (**E**,**J**,**O**)). In images with a score of 3, note the cranial cruciate ligament (black arrow), caudal cruciate ligament (white arrow), meniscofemoral ligament (*), medial meniscus (black arrowhead), and lateral meniscus (white arrowhead).

**Figure 3 animals-14-03334-f003:**
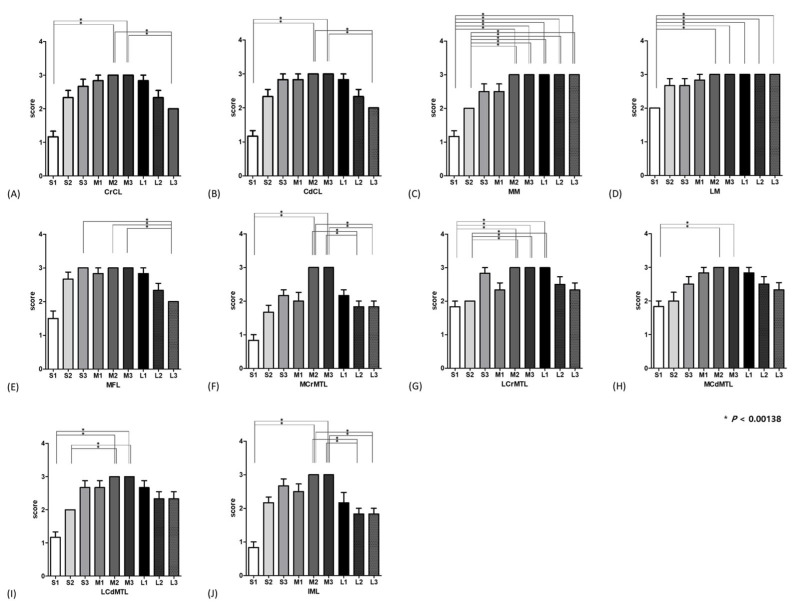
The graphs illustrate the average visualization scores for the ligamentous structures and menisci of the stifle during extension in each group. This includes: (**A**) cranial cruciate ligament, (**B**) caudal cruciate ligament, (**C**) medial meniscus, (**D**) lateral meniscus, (**E**) meniscofemoral ligament, (**F**) medial cranial meniscotibial ligament, (**G**) lateral cranial meniscotibial ligament, (**H**) medial caudal meniscotibial ligament, (**I**) lateral caudal meniscotibial ligament, (**J**) intermeniscal ligament.

**Table 1 animals-14-03334-t001:** The group of CT arthrography protocols.

Group	Concentration of Contrast Medium (mgI/mL)	Injection Volume (mL/kg)
S1	50	0.2
S2	50	0.3
S3	50	0.4
M1	100	0.2
M2	100	0.3
M3	100	0.4
L1	150	0.2
L2	150	0.3
L3	150	0.4

**Table 2 animals-14-03334-t002:** Scoring standard for assessing the quality of visualization of stifle ligamentous structures and menisci.

Score	Findings
0	The structure was not visible
1	The structure was poorly visualized but detectable, identified by its location and filling defect, not by margins or shape
2	The structure was visualized based on its location, shape, and filling defect, but the margins were not clearly defined due to underfilling or artifacts.
3	The structure was well visualized and clearly delineated, characterized by its location, shape, filling defect, and margins.

**Table 3 animals-14-03334-t003:** The intra- and inter-observer agreement: Fleiss’ kappa with 95% confidence intervals among four observers for the assessment of visualization scores of stifle joint ligamentous structures and menisci.

Variables	Intra-Observer Agreement	Inter-Observer Agreement
CrCL	0.702 (0.577 to 0.827)	0.707 (0.614 to 0.800)
CdCL	0.681 (0.557 to 0.805)	0.746 (0.653 to 0.838)
MM	0.680 (0.566 to 0.794)	0.650 (0.563 to 0.737)
LM	0.639 (0.501 to 0.777)	0.684 (0.589 to 0.779)
MFL	0.621 (0.499 to 0.744)	0.574 (0.484 to 0.663)
MCrMTL	0.434 (0.326 to 0.542)	0.450 (0.372 to 0.528)
LCrMTL	0.524 (0.415 to 0.634)	0.458 (0.377 to 0.539)
MCdMTL	0.536 (0.422 to 0.651)	0.537 (0.454 to 0.619)
LCdMTL	0.597 (0.483 to 0.711)	0.547 (0.465 to 0.629)
IML	0.544 (0.439 to 0.650)	0.515 (0.439 to 0.592)

CrCL, cranial cruciate ligament; CdCL, caudal cruciate ligament; MM, medial meniscus; LM, lateral meniscus; MFL, meniscofemoral ligament; MCrMTL, medial cranial meniscotibial ligament; LCrMTL, lateral cranial meniscotibial ligament; MCdMTL, medial caudal meniscotibial ligament; LCdMTL, lateral caudal meniscotibial ligament; IML, intermeniscal ligament.

## Data Availability

The data present in this study are available within the article.

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
