# Peer review of "Optimal Computed Tomographic Arthrography Protocol for Stifle Ligamentous Structure and Menisci in Dogs"

_animals, 2024, doi:10.3390/ani14223334_

Round 1
Reviewer 1 Report
Comments and Suggestions for Authors
Dear authors,
Congratulations on completing your research. I have a few queries if you could answer them please and deal with them in a revision:
1. Please elaborate on the ethical approval for this research, as it would be unusual to do repeated procedures on the same animal?
2. Is their an error on Table 1? The volumes look different to that described in the text?
3. You mention veterinary radiologist. Please define veterinary radiologist exactly e.g. what specialist qualifications and experience.
4. Why remove the contrast from the joint after the arthrography?
5. The numbers are low so how reliable could the statistics be? Please provide an analysis of this. Can we really make statistical conclusions with such low numbers?
6. You mention patient positioning as a critical factor, please elaborate with reference to your chosen position.
7. You mention the reference 13 which did similar research on cadavers. Please do an extensive compare and contrast analysis between this paper and your paper
8. What does your paper add or change in the literature ie. its significance.
Yours sincerely
Reviewer.
6.
Author Response
Reviewer 1
Congratulations on completing your research. I have a few queries if you could answer them please and deal with them in a revision:
â–º We sincerely thank you for your efforts in reviewing our manuscript. We have included our point-by-point responses to your comments below.
- Please elaborate on the ethical approval for this research, as it would be unusual to do repeated procedures on the same animal?
â–º We appreciate your comment. I completely agree with the reviewer’s comments. Due to the limited number of animals involved in the study, we aimed to increase the reliability and accuracy of the research by applying all nine protocols to the same subjects whenever possible. To ensure ethical standards for animal welfare, we incorporated a minimum two-week rest period between experiments. During this time, we closely monitored the animals' conditions and conducted thorough assessments.
- Is their an error on Table 1? The volumes look different to that described in the text?
â–º We appreciate your comment. We apologize for the error in our table. We have corrected the volume in Table 1 to match the content in the main text.
- You mention veterinary radiologist. Please define veterinary radiologist exactly e.g. what specialist qualifications and experience.
â–º We appreciate your comment. We have revised "radiologist" to "veterinarian" in line with the reviewer’s suggestion and updated the description to “experienced veterinary veterinarian with over 3 years of experience in radiology.” (Line 128- 137)
- Why remove the contrast from the joint after the arthrography?
â–º We appreciate your comment. The presence of contrast agent in the joint can lead to discomfort and potentially affect subsequent arthrography procedures. Given the likelihood of post-procedural swelling and pain, the contrast agent was removed through arthrocentesis. This has been added in the following sentence.
Line 125-126: “Once the CT imaging was completed, the contrast agent was removed through arthrocentesis to reduce swelling, pain, and potential impact on subsequent arthrography procedures.”
- The numbers are low so how reliable could the statistics be? Please provide an analysis of this. Can we really make statistical conclusions with such low numbers?
â–º We appreciate your comment. I completely agree with the reviewer’s comments. Since the statistical power may be low, we attempted to conduct all nine protocols on the same subjects as much as possible. We have now added a mention in the limitations section regarding the potential for statistical errors due to the small number of animals, emphasizing the need for further studies with a larger sample size. Additionally, we are currently conducting a retrospective study of CTA in numerous patients suspected of having various ligamentous diseases in the joints.
Line 328-330: “Additionally, due to the small number of animals, there is a possibility that statistical Type I and Type II errors may have occurred, which underscores the need for further studies involving a larger number of animals.”
- You mention patient positioning as a critical factor, please elaborate with reference to your chosen position.
â–º We appreciate your comment. There are ligaments in the stifle joint that may be better visualized depending on the angle. Previous studies have indicated that the cranial cruciate ligament (CCL) is best visualized at a 90-degree angle. However, in this study, we aimed to evaluate various ligaments and cartilage, so imaging was performed at a 135-degree angle, which is the most natural standing position. We have also mentioned that further evaluation of differences at various angles will be necessary in future studies.
Line 336-340: “Lastly, because imaging at flexion angles was not feasible, a comparative analysis using different extension positions could not be performed. This limitation means that the ideal positioning for CTA remains unverified, highlighting the need for additional research to determine the optimal imaging posture for CTA.”
- You mention the reference 13 which did similar research on cadavers. Please do an extensive compare and contrast analysis between this paper and your paper
â–º We appreciate your comment. Reference 13 is a study where the contrast agent volume was consistently set at 1.5 mL, while the concentration was adjusted to 300, 150, and 75 mg/mL. Consequently, this differs from our study, as appropriate contrast agent volume may vary with concentration. Additionally, the concentrations of 300 mgI/mL and 150 mgI/mL may be relatively high for arthrography. In our study, we aimed to determine the appropriate volumes at lower concentrations of 150, 100, and 50 mgI/mL. Based on this, we added the following sentence:
Line 240-244: “However, this study did not consider that ligament visibility may improve if volume is adjusted based on concentration, as it consistently used a volume of 1.5 mL regardless of concentration. Moreover, concentrations of 300 mgI/mL and 150 mg/mL are relatively high compared to other studies on joint arthrography, highlighting the need for further research on lower contrast agent concentrations.”
- What does your paper add or change in the literature ie. its significance.
â–º We appreciate your comment. There has been a lack of studies that simultaneously adjust both the volume and concentration to determine the optimal CT arthrography protocol. To further emphasize its importance, as suggested, we added the following sentence to the conclusion:
Line 342-343: “Identifying the optimal concentration and volume of contrast agent is essential to enhance ligament visibility in CT arthrography while minimizing adverse effects.”

Reviewer 2 Report
Comments and Suggestions for Authors
Comments and Suggestions for Authors:
Congratulations on your research. The paper is well-executed. I have a few suggestions to improve the clarity of the text:
Lines 94: In section 2.2, it should be clarified whether the 9 proposed protocols were performed on each patient. The total number of studies conducted is not indicated. If all 9 protocols were not performed on the 6 animals in the study, please provide a justification
Lines 114: In Table 1, the injection volume (mL/kg) must be corrected for the following protocols: S2 (replace 0.2 to 0.3), S3 (replace 0.2 to 0.4), M1 (replace 0.3 to 0.2), M3 (replace 0.3 to 0.4), L1 (replace 0.4 to 0.2) and L2 (replace 0.4 to 0.3)
Line 131: Replace “Before interpreting the images, the radiologists received instructional materials that included a reference paper13 and six normal canine CT stifle arthrograms” with “Before interpreting the images, the radiologists received instructional materials that included a reference paper [13] and six normal canine CT stifle arthrograms”
Lines 135-139: In section 2.4, it would be helpful to include a figure showing the transverse, sagittal, and coronal planes of the canine stifle to improve clarity.
Lines 140-144: Consider adding a figure showing the knee structures that can be evaluated in CT arthrography. Image "O" in Figure 1 of the article is too small.
Line 193: Increase the size of the graphs shown in Figure 2, as they are currently too small.
Lines 272-273: Replace “The medial meniscus is firmly attached to the tibial posterior intercondylar area and is located in front of the posterior cruciate ligament” with “The medial meniscus is firmly attached to the tibial posterior intercondylar area and is located in front of the caudal cruciate ligament”
Lines 217-319: In section 4 (Discussion), to discuss the possible differences between your study and similar studies referenced in the introduction, specifically regarding the number of slices used in the CT scanner. Specifically, with reference no. 13 shown on lines 229-232.
Lines 403-406. The DOI for reference no. 26 is 10.3348/kjr.2014.15.6.739
Lines 418-419. The DOI for reference no. 32 is 10.3348/kjr.2008.9.6.520
Author Response
Reviewer 2
Congratulations on your research. The paper is well-executed. I have a few suggestions to improve the clarity of the text:
â–º We sincerely thank you for your efforts in reviewing our manuscript. We have included our point-by-point responses to your comments below.
Lines 94: In section 2.2, it should be clarified whether the 9 proposed protocols were performed on each patient. The total number of studies conducted is not indicated. If all 9 protocols were not performed on the 6 animals in the study, please provide a justification
â–º We appreciate your comment. All nine protocols were performed on the six animals that participated in the study. For clarity, I have revised the following sentence in section 2.3.
Line 104-106: “A total of 54 procedures were performed in six dogs, with each dog undergoing the Nine CT arthrography protocols using various concentrations and volumes of contrast agents.”
Lines 114: In Table 1, the injection volume (mL/kg) must be corrected for the following protocols: S2 (replace 0.2 to 0.3), S3 (replace 0.2 to 0.4), M1 (replace 0.3 to 0.2), M3 (replace 0.3 to 0.4), L1 (replace 0.4 to 0.2) and L2 (replace 0.4 to 0.3)
â–º We appreciate your comment. We apologize for the error in our table. We have corrected the volume in Table 1 to match the content in the main text.(Table 1)
Line 131: Replace “Before interpreting the images, the radiologists received instructional materials that included a reference paper13 and six normal canine CT stifle arthrograms” with “Before interpreting the images, the radiologists received instructional materials that included a reference paper [13] and six normal canine CT stifle arthrograms”
â–º We appreciate your comment. As you suggested, I have revised the sentence to: 'Before interpreting the images, the radiologists received instructional materials that included a reference paper [13] and six normal canine CT stifle arthrograms. (Line 131)
Lines 135-139: In section 2.4, it would be helpful to include a figure showing the transverse, sagittal, and coronal planes of the canine stifle to improve clarity.
â–º We appreciate your comment. We have added Figure 1, which displays all ligaments visible in the sagittal, axial, and coronal planes (Line 142, Figure 1)
Figure 1. Visualization of stifle joint ligamentous structures and menisci in CT arthrography. Axial (A), coronal (B and C), and sagittal (D-I) planes. cranial cruciate ligament (a), caudal cruciate lig-ament (b), medial meniscus (c), lateral meniscus (d), meniscofemoral ligament (e), medial cranial me-niscotibial ligament (f), lateral cranial meniscotibial ligament (g), medial caudal meniscotibial ligament (h), lateral caudal meniscotibial ligament (i).
Lines 140-144: Consider adding a figure showing the knee structures that can be evaluated in CT arthrography. Image "O" in Figure 1 of the article is too small.
â–º We appreciate your comment. We have adjusted Figure 2O by enlarging the image.
Line 193: Increase the size of the graphs shown in Figure 2, as they are currently too small.
â–º We appreciate your comment. We have increased the size of the graph and added it.
Lines 272-273: Replace “The medial meniscus is firmly attached to the tibial posterior intercondylar area and is located in front of the posterior cruciate ligament” with “The medial meniscus is firmly attached to the tibial posterior intercondylar area and is located in front of the caudal cruciate ligament”
â–º We appreciate your comment. I have revised it to: 'The medial meniscus is firmly attached to the tibial posterior intercondylar area and is located in front of the caudal cruciate ligament.' (Line 291)
Lines 217-319: In section 4 (Discussion), to discuss the possible differences between your study and similar studies referenced in the introduction, specifically regarding the number of slices used in the CT scanner. Specifically, with reference no. 13 shown on lines 229-232.
â–º We appreciate your comment. I fully agree with the reviewer’s comments. Previously, the various ligaments may not have been clearly visualized due to the use of thick CT slices. And reference 13 is a study where the contrast agent volume was consistently set at 1.5 mL, while the concentration was adjusted to 300, 150, and 75 mg/mL. Consequently, this differs from our study, as appropriate contrast agent volume may vary with concentration. Additionally, the concentrations of 300 mgI/mL and 150 mgI/mL may be relatively high for arthrography. In our study, we aimed to determine the appropriate volumes at lower concentrations of 150, 100, and 50 mgI/mL. Based on this, we added the following sentence:
Line 240-245: “However, this study did not consider that ligament visibility may improve if volume is adjusted based on concentration, as it consistently used a volume of 1.5 mL regardless of concentration. Moreover, concentrations of 300 mgI/mL and 150 mg/mL are relatively high compared to other studies on joint arthrography, highlighting the need for further research on lower contrast agent concentrations.”
Lines 403-406. The DOI for reference no. 26 is 10.3348/kjr.2014.15.6.739
â–º We appreciate your comment. I have made the revisions.
Lines 418-419. The DOI for reference no. 32 is 10.3348/kjr.2008.9.6.520
â–º We appreciate your comment. I have made the revisions.

Reviewer 3 Report
Comments and Suggestions for Authors
Dear authors, thank you for submitting this interesting manuscript. Your study, aimed at optimizing the concentration and injection volume of the contrast medium to improve evaluation of the stifle joint's ligamentous structures and menisci, is well structured and detailed.
In my opinion, your study makes an important contribution to the understanding of the advantages and disadvantages of computed tomographic arthrography (CTA) and may help to standardise CTA protocols. The clarity of presentation makes the manuscript a stimulating and useful read for both researchers and clinicians in the field. Congratulations for your work.
I have only few suggestions for your work:
Line 71-72 “while a high concentration may result in better tissue differentiation but increase the occurrence of bloom artifacts [13]” Regarding the intra-articular use of iodinated contrast media, it would be appropriate to briefly discuss the associated risks due to the chondrotoxic effect of these substances.
Line 85-86 “Prior to receiving general anesthesia, all dogs underwent detailed physical examinations and thoracic and musculoskeletal radiographic imaging.” Were a blood sample for haematobiochemical analysis and an anaesthesia examination performed?
Line 88-89 “Additionally, as part of the procedure, the dogs were given a two-week break period” I cannot understand the purpose of the two-week break period granted to the study subjects. Could you please provide further explanation?
Line 106-110 “A 50mgI/mL concentration of contrast medium was injected at volumes of 0.2mL/kg, 0.3mL/kg, and 0.4mL/kg (protocols S1, S2, and S3). A 100mgI/mL concentration of contrast medium was injected at volumes of 0.2mL/kg, 0.3mL/kg, and 0.4mL/kg (protocols M1, M2, and M3). A 150mgI/mL concentration of contrast medium was injected at volumes of 0.2mL/kg, 0.3mL/kg, and 0.4mL/kg (protocols L1, L2, and L3).” I think that what is described is not correctly reported in Table 1.
Line124-125 “Once the CT imaging was completed, the contrast agent was removed through arthrocentesis.” Were the joints washed to ensure that any residual contrast was thoroughly removed?
Line 243-244: “In addition, hypertonic iodinated solutions have been associated with post-procedural swelling and pain [29].” See Line 71-72
Author Response
Reviewer 3
Dear authors, thank you for submitting this interesting manuscript. Your study, aimed at optimizing the concentration and injection volume of the contrast medium to improve evaluation of the stifle joint's ligamentous structures and menisci, is well structured and detailed.
In my opinion, your study makes an important contribution to the understanding of the advantages and disadvantages of computed tomographic arthrography (CTA) and may help to standardise CTA protocols. The clarity of presentation makes the manuscript a stimulating and useful read for both researchers and clinicians in the field. Congratulations for your work.
I have only few suggestions for your work:
â–º We sincerely thank you for your efforts in reviewing our manuscript. We have included our point-by-point responses to your comments below.
Line 71-72 “while a high concentration may result in better tissue differentiation but increase the occurrence of bloom artifacts [13]” Regarding the intra-articular use of iodinated contrast media, it would be appropriate to briefly discuss the associated risks due to the chondrotoxic effect of these substances.
â–º We appreciate your comment. As mentioned above, we have added the following sentence to the discussion section.
Line 256-263: “High-concentration contrast agents can alter the osmotic pressure within cartilage tissue, potentially increasing water content and inducing swelling. Additionally, high-concentration iodine contrast agents have an increased propensity to diffuse into damaged cartilage regions by following micro-cracks on the surface. This enhanced penetration improves lesion visualization but also elevates the risk of inflammation and pain due to localized edema and tissue reaction in structurally compromised cartilage.”
Line 85-86 “Prior to receiving general anesthesia, all dogs underwent detailed physical examinations and thoracic and musculoskeletal radiographic imaging.” Were a blood sample for haematobiochemical analysis and an anaesthesia examination performed?
â–º We appreciate your comment. It seems we missed it by mistake. Haematobiochemical analysis was also conducted, and we have added this information to the main text. (Line 86)
Line 88-89 “Additionally, as part of the procedure, the dogs were given a two-week break period” I cannot understand the purpose of the two-week break period granted to the study subjects. Could you please provide further explanation?
â–º We appreciate your comment. To minimize the effects of previous arthrography and allow for recovery from repeated anesthesia, we conducted the procedures with a two-week interval.
Line 106-110 “A 50mgI/mL concentration of contrast medium was injected at volumes of 0.2mL/kg, 0.3mL/kg, and 0.4mL/kg (protocols S1, S2, and S3). A 100mgI/mL concentration of contrast medium was injected at volumes of 0.2mL/kg, 0.3mL/kg, and 0.4mL/kg (protocols M1, M2, and M3). A 150mgI/mL concentration of contrast medium was injected at volumes of 0.2mL/kg, 0.3mL/kg, and 0.4mL/kg (protocols L1, L2, and L3).” I think that what is described is not correctly reported in Table 1.
â–º We appreciate your comment. We apologize for the error in our table. We have corrected the volume in Table 1 to match the content in the main text. (table 1)
Line124-125 “Once the CT imaging was completed, the contrast agent was removed through arthrocentesis.” Were the joints washed to ensure that any residual contrast was thoroughly removed?
â–º We appreciate your comment. If the contrast agent remains in the joint, it can cause discomfort or affect subsequent arthrography, and there may be side effects such as swelling or pain after the procedure. Therefore, we made every effort to remove it as much as possible. Previous studies have also indicated that while efforts were made to remove the contrast agent to the greatest extent possible, complete removal is deemed impossible. However, we allowed a two-week interval and confirmed the absence of the contrast agent in the pre-contrast CT before proceeding with the arthrography.
Line 243-244: “In addition, hypertonic iodinated solutions have been associated with post-procedural swelling and pain [29].” See Line 71-72
â–º We appreciate your comment. As mentioned above, we have added the following sentence to the discussion section.
“High-concentration contrast agents can alter the osmotic pressure within cartilage tissue, potentially increasing water content and inducing swelling. Additionally, high-concentration iodine contrast agents have an increased propensity to diffuse into damaged cartilage regions by following micro-cracks on the surface. This enhanced penetration improves lesion visualization but also elevates the risk of inflammation and pain due to localized edema and tissue reaction in structurally compromised cartilage.”
